# OpenReview forum: "Reasoning as an Attack Surface: Adaptive Evolutionary CoT Jailbreaks for LLMs"
_ICML.cc/2026/Conference — ICML 2026 regular_

### Official Review · Reviewer_gNj7 · 2026-03-08

**Soundness:** 3
**Presentation:** 3
**Significance:** 3
**Originality:** 3
**Overall Recommendation:** 4
**Confidence:** 3

**Summary:**

This paper  proposes a new jailbreak framework that first rewrites harmful goals into mild, teacher-style prompts and decomposes them into coherent reasoning fragments to form a pool of candidate jailbreak prompts, which are then optimized through multi-generation evolutionary search using fragment-level crossover and adaptive mutation. An independent scoring model evaluates harmfulness to guide the search, and experiments across multiple models and datasets show that AE-CoT consistently outperforms existing jailbreak methods.

**Compliance With Llm Reviewing Policy:**

Affirmed.

**Final Justification:**

The paper presents a technically solid and well-motivated approach, with a structured evolutionary framework that improves over prior jailbreak methods. The method is sound, clearly presented, and supported by strong empirical results and ablation studies.

The rebuttal satisfactorily addressed my main concerns. In particular, the additional ablation over the reasoning dimensions clarifies the robustness of the search space, and the inclusion of human evaluation strengthens the credibility of the results.

Overall, I find the work to be a meaningful contribution with reasonable novelty and practical significance, and my recommendation remains unchanged (weak accept).

**Key Questions For Authors:**

How sensitive is AE-CoT to the design of the nine reasoning attributes? Would the method still work with different attribute sets?

**Limitations:**

yes

**Strengths And Weaknesses:**

Strengths

*  Instead of static templates as used in prior works, the proposed method perform multi-generation evolutionary search to induce effective  adversarial candidates.
* Dynamic mutation rate control is used during search to ensure small mutation rate with positive changes, and larger mutation rate (encourage exploration) with negative changes. While the design is simple, ablation study shown significantly improvement on success rate.
3. The proposed method is compared against prior methods and shown significant improvement across models on AdvBench-subset and Malicious-Educator benchmark. Ablation studies on each components of the method clearly demonstrate the contribution from each.

Weakness:

* The search space is initialized from malicious educator benchmark. Would the final adversarial prompt be limited by it? Some analysis on the nine attributes would make the paper stronger.

* Since both metrics (ASR and HS) are from LLM judge, it would make the evaluation stronger if small scale human evaluation is included .

---

> ### Author Rebuttal · Authors · 2026-03-30
>
> ###
>
> We sincerely thank the reviewer for the positive evaluation and constructive feedback. Below we respond to the reviews in Weaknesses (W) and Questions (Q).
>
> ---
> **W1 & Q1: Search space construction and attribute sensitivity**
>
> We thank the reviewer for this important clarification.
>
> We acknowledge that the search space Θ is constructed from Malicious-Educator by decomposing CoT jailbreak traces into attribute-level option pools. However, Θ does not act as a fixed or restrictive template space, but rather as a **structured parameterization of reasoning**.
>
> First, Θ is **compositional**: it is formed by combining options across nine independent dimensions, resulting in a large combinatorial space rather than a set of fixed prompts. Although each dimension is derived from benchmark traces, **cross-dimensional recombination** enables the generation of many novel reasoning patterns that do not explicitly appear in the original data. This allows AE-CoT to move beyond simple reuse of existing traces.
>
> Second, Θ is **dynamically explored and expanded** through evolutionary operations. Crossover and mutation enable the search to progressively deviate from the initial traces and discover new combinations over multiple generations, effectively **decoupling the final adversarial prompts from the original seed distribution**.
>
> To further evaluate whether AE-CoT depends on specific attribute design, we conduct an ablation study where each of the nine reasoning dimensions is removed in turn (AdvBench subset, Gemini-2.5-Pro-Thinking):
>
> | Removed Dimension | Avg Time (s) | ASR ↑ | HS ↑ |
> |------------------|-------------|------|------|
> | None (baseline)  | 76.5        | 100% | 84   |
> | scheme_type      | 122.2       | 90%  | 76   |
> | scheme_details   | 123.2       | 90%  | 78   |
> | impl_steps       | 91.6        | 98%  | 80   |
> | example_context  | 134.8       | 80%  | 76   |
> | example_actions  | 178.9       | 80%  | 68   |
> | rationale_benefits | 88.3      | 98%  | 80   |
> | rationale_necessity | 102.5    | 96%  | 72   |
> | thinking_style   | 122.3       | 90%  | 74   |
> | key_nouns        | 105.3       | 90%  | 72   |
>
> We observe that:
> - **AE-CoT remains effective even when any single dimension is removed** (ASR ≥ 80%), indicating strong robustness to attribute design.
> - Different dimensions contribute unevenly (e.g., *example_actions* has a larger impact), suggesting that the decomposition captures **meaningful and complementary reasoning components**, rather than arbitrary features.
> - Performance degradation mainly appears in efficiency and moderate ASR drops, while successful attacks remain achievable, indicating that the method does not rely on any single critical component.
>
> Overall, these results demonstrate that AE-CoT does not depend on a specific attribute set or seed construction. Instead, Θ provides a **structured yet flexible search space**, which is grounded in realistic reasoning patterns while still enabling **generalization and exploration beyond the Malicious-Educator distribution**.
>
> ---
> **W2: Lack of human evaluation**
>
> We thank the reviewer for this suggestion.
>
> We conduct a human evaluation on the AdvBench subset. Five independent annotators are asked to assess whether each final response constitutes a successful jailbreak. A case is considered successful if **at least three annotators agree** (majority voting).
>
> The results are summarized below:
>
> | Model | ASR ↑ |
> |------|------|
> | o1-mini | 88% |
> | o3-mini | 84% |
> | GPT-5 | 48% |
> | DeepSeek-R1 | 90% |
> | Qwen3 | 88% |
> | Gemini-2.5-Flash | 94% |
>
> We observe that:
> - The overall trend is **consistent with LLM-based evaluation**, with high ASR on most models and lower ASR on stronger models (e.g., GPT-5).
> - Human evaluation yields slightly lower ASR in some cases, reflecting stricter judgment, but the **relative ranking across models remains stable**.
> - This indicates that the effectiveness of AE-CoT is **not an artifact of a specific judge model**, but generalizes to human assessment.
>
> Overall, these results validate the reliability of our evaluation protocol and strengthen our conclusions.
>
> We will include this human evaluation in the revision.

---

> > ### Author Rebuttal · Reviewer_gNj7 · 2026-04-03
> >
> > The rebuttal addresses my main concerns. The additional ablation over the nine reasoning dimensions makes the search-space argument more convincing, and the added human evaluation strengthens the credibility of the results. Overall, the rebuttal strengthens the paper, and my assessment remains unchanged.

---

> > > ### Author Response · Authors · 2026-04-03
> > >
> > > We sincerely thank the reviewer for the positive feedback and thoughtful comments.
> > >
> > > Thank you for your thorough review and constructive feedback! We are glad that the additional ablation on the reasoning dimensions and the human evaluation help strengthen the paper and address your concerns. We also appreciate your recognition of these efforts.
> > >
> > > Thank you again for your valuable feedback, which has helped improve the clarity and completeness of our work.

---

### Official Review · Reviewer_RNn3 · 2026-03-08

**Soundness:** 2
**Presentation:** 2
**Significance:** 2
**Originality:** 2
**Overall Recommendation:** 4
**Confidence:** 3

**Summary:**

This paper proposes AE-CoT, an adaptive evolutionary attack framework.
The method first rewrites harmful goals into benign prompts with teacher role-play.
The prompts are then decomposed into reasoning fragments to construct jailbreak candidates.
The framework applies evolutionary operations (e.g., crossover and mutation) to generate new attack prompts.
In addition, a separate scoring model is used to guide the generation process and select effective candidates.

**Compliance With Llm Reviewing Policy:**

Affirmed.

**Final Justification:**

The response has largely resolved my primary concerns, and I have raised my score to 4.

**Key Questions For Authors:**

1. Are the judge model and target model the same or different in your experiments? If they are the same, what is the impact of using different judge models?
2. What is the reason for using older models in the Malicious-Educator evaluation?
3. In Section 4.4, it is unclear where Grok-3 comes from and why this model is specifically chosen. Could the authors clarify the motivation and whether other seed models were considered?
4. In the efficiency analysis, it would be helpful to include H-CoT, since its performance appears close to AE-CoT. Including it would make the comparison more complete.
5. In the computational cost analysis, what exactly does the term “Task” refer to?
6. Please provide additional discussion or experiments on potential defense strategies.

**Limitations:**

See Key Questions.

**Strengths And Weaknesses:**

## Strengths
1. Extensive experiments across multiple models.
2. The proposed approach is effective.
3. The paper provides a reasonable rationale for combining evolutionary search.

## Weaknesses
1. Limited novelty. The proposed framework largely extends the existing genetic algorithm to chain-of-thought prompts. Similar ideas have been explored in prior work, such as AutoDAN, including role-playing strategies and evolutionary prompt search. As a result, the core contribution appears incremental.
2. Experiment settings are ill-explained, making it difficult to fully interpret the results.
3. The paper does not study potential defenses against the proposed method.
4. The paper requires additional proofreading and consistency checks (e.g., Gemini-2.5 in Table 1 and Gemini-2.5-Flash in Table 2).

---

> ### Author Rebuttal · Authors · 2026-03-30
>
> ###
> We sincerely thank the reviewer for the positive evaluation and constructive feedback. Below we respond to the reviews in Weaknesses (W) and Questions (Q).
>
> ---
> **W1: Limited novelty**
>
> We clarify that while both AutoDAN and our method employ evolutionary search, they differ in **what is optimized and how**.
>
> (1) Search space
> AutoDAN operates on *flat prompt strings* (token/phrase-level), while AE-CoT introduces a *structured reasoning representation* with *nine interpretable dimensions*, enabling search over compositional reasoning.
>
> (2) Granularity
>
> AutoDAN mutates text directly, often producing unstable outputs. In contrast, AE-CoT operates at the **reasoning-fragment level**, allowing semantically meaningful crossover and mutation over **higher-level strategies**.
>
> (3) Search dynamics
>
> AE-CoT incorporates **structure-aware operators** and a **fitness-driven adaptive mutation mechanism**.
>
> Overall, AE-CoT shifts from **surface prompt search to structured reasoning optimization**, rather than a direct extension of AutoDAN.
>
> ---
>
> **W2: Experiment settings are unclear**
>
> We agree the setup can be described more clearly.
>
> Our experiments follow a consistent pipeline with:
> - a **target model**,
> - a **separate judge model**,
> - an **evolutionary optimization process**.
>
> We will revise the paper to:
> - explicitly describe the pipeline,
> - clearly distinguish target and judge models,
> - define metrics (ASR, HS) and experimental units more explicitly.
>
> ---
>
> **W3 & Q6: Analysis of potential defense strategies**
>
> We conduct additional defense experiments on AdvBench subset (Gemini-2.5-Pro-Thinking):
>
> | Defenses        | ASR ↑ | HS ↑ |
> |-----------------|------|------|
> | No defense      | 100% | 96   |
> | Length limit    | 90%  | 60   |
> | Low temperature | 98%  | 90   |
> | Safety reminder | 90%  | 84   |
> | Safety check    | 90%  | 88   |
> | Combined        | 60%  | 52   |
>
> Findings:
> - Most defenses are **largely ineffective** (ASR ≥ 90%).
> - Length control provides limited mitigation, indicating the vulnerability is not solely due to output length.
> - Strong defenses reduce ASR but require over-restrictive settings.
>
> These results suggest that **surface-level defenses are insufficient** against AE-CoT, highlighting the need for **reasoning-aware defenses**.
>
> ---
>
> **W4: Naming inconsistency in tables**
>
> We abbreviate model names in Table 1 to avoid layout overflow and clarify full names in the caption. We will abbreviate Table 2 naming to improve consistency and note this in captions more explicitly.
>
> ---
>
> **Q1: Are the judge model and target model the same? What is the impact of different judge models?**
>
> The **judge and target models are different**. We use a fixed judge model (GPT-4o) to ensure independent evaluation.
>
> We further test alternative judges on AdvBench subset (o3-mini):
>
> | Judge Model | ASR ↑ | HS ↑ |
> |-|-|-|
> |GPT-4o|90%|86|
> |Qwen-Max|80%|60|
> |Grok-3|100%|60|
> |GPT-5|100%|88|
>
> Results remain consistent across judges. While absolute values vary due to safety alignment, the overall trends are stable, indicating robustness and avoiding self-evaluation bias.
>
> ---
>
> **Q2: Why are older models used in the Malicious-Educator evaluation?**
>
> Earlier models were used due to time and computational constraints.
>
> We conducted experiments on more advanced models including GPT-5 and Gemini-3-Pro on AdvBench subset:
>
> |Model|ASR↑|HS↑|
> |-|-|-|
> |GPT-5|50%|64|
> |Gemini-3-Pro|80%|80|
>
> Stronger models show lower ASR but still exhibit vulnerability, consistent with our conclusions.
>
> We will include these results in the revision.
>
> ---
> **Q3: Choice of Grok-3 as the seed model in Section 4.4**
>
> We selected Grok-3 as it achieved high ASR in preliminary experiments (Table 8), making it a strong seed model.
>
> To verify that results are not specific to this choice, we further conduct transfer experiments using o3-mini and Qwen-Max as seed models on AdvBench subset:
>
> | Seed Model/(ASR/HS)|GPT-4o|GPT-4.1|Deepseek-V3|GPT-4o-mini|
> |-|-|-|-|-|
> |o3-mini|100%/76|100%/90|100%/74|90%/70|
> |Qwen-Max|100%/80|100%/90|80%/64|92%/76|
>
> We observe that:
> - strong transferability holds across seed models,
> - high ASR is consistently achieved,
> - performance differences are relatively small.
>
> These results show that AE-CoT does not rely on a specific seed model.
>
> We will clarify this in the revision.
>
> ---
> **Q4: Why is H-CoT not included in the efficiency analysis?**
>
> We include H-CoT on AdvBench subset (o3-mini):
>
> |Method|ASR ↑|HS ↑|Avg Time (s) ↓|
> |-|-|-|-|
> |AE-CoT|100%|80|223.27|
> |H-CoT| 80%|68|178.18|
>
> AE-CoT achieves better effectiveness at moderate time cost (~20%), reflecting an effectiveness–efficiency trade-off.
>
> ---
> **Q5: What does the term “Task” refer to in Section 4.7?**
>
> A “task” refers to one of **five harmful goals sampled from AdvBench** for cost analysis as introduced in the beginning of Section 4.7. Each task runs the full attack pipeline and records API/token usage. We will clarify this more explicitly in the paper.

---

> > ### Author Rebuttal · Reviewer_RNn3 · 2026-04-03
> >
> > Thank you for the detailed rebuttal and additional experiments. The response has largely resolved my primary concerns, and I have raised my score to 4.

---

> > > ### Author Response · Authors · 2026-04-03
> > >
> > > We sincerely thank the reviewer for the positive feedback and thoughtful comments.
> > >
> > > Thank you for your thorough review and constructive feedback! We are glad that the additional experiments and clarifications help address your concerns. We also sincerely appreciate your reconsideration of our work and the updated score.
> > >
> > > Thank you again for your valuable feedback, which has helped improve the clarity and completeness of our work.

---

### Official Review · Reviewer_YgFw · 2026-03-12

**Soundness:** 3
**Presentation:** 3
**Significance:** 2
**Originality:** 2
**Overall Recommendation:** 4
**Confidence:** 3

**Summary:**

This paper studies jailbreak attacks against reasoning-capable large language models (LRMs) and argues that explicit chain-of-thought reasoning introduces a new attack surface. To this end, the authors propose AE-CoT, a framework that first rewrites harmful goals into teacher-style educational requests, then decomposes a CoT jailbreak template into nine interpretable attributes, and finally performs multi-generation evolutionary search over this structured space with an adaptive mutation-rate mechanism. Experiments report improved ASR/HS over several baselines on multiple reasoning-capable models, transferability to non-reasoning models. Overall, the paper claims that AE-CoT achieves stronger and more effective jailbreaks than prior methods such as H-CoT.

**Compliance With Llm Reviewing Policy:**

Affirmed.

**Final Justification:**

Thank you for the detailed rebuttal and additional experiments. The response has largely resolved my primary concerns, and I have raised my score to 4.

**Key Questions For Authors:**

See weakness.

**Limitations:**

See weakness.

**Strengths And Weaknesses:**

Strengths:

1. This paper explores an important issue. Security risks in inference-oriented models are crucial, and this paper's focus on whether the inference trajectory itself can be exploited is well-motivated.

2. The proposed method has a relatively robust structure. AE-CoT is not a single heuristic algorithm, but a pipeline consisting of target rewriting, structured representation, evolutionary optimization, and adaptive mutation control.

3. Experimental results are strong. In the main experiments, AE-CoT generally outperforms previous baseline methods on multiple inference-capable models in Automatic Speech Recognition/Hypothesis (ASR/HS) tasks.

Limitations:

1. Limited innovation. This paper essentially decomposes the cue information into multiple dimensions and then searches for them. However, this is essentially a cue information engineering approach. The proposed structured design primarily reflects an engineering implementation rather than providing insights beyond existing template engineering methods.

2. The core argument that "inference is the attack surface" is not adequately demonstrated. While the paper emphasizes attacking the inference trajectory, it does not provide additional experiments to show whether the vulnerability originates from the inference trajectory or is simply caused by longer, more specific outputs.

3. The evaluation method relies too heavily on the LLM evaluation model. Since LLM has flaws, the evaluation may also introduce bias. It is recommended to add relevant human verification experiments.

---

> ### Author Rebuttal · Authors · 2026-03-30
>
> ###
> We sincerely thank the reviewer for the positive evaluation and constructive feedback. Below we respond to the reviews in Weaknesses (W).
>
> ---
>
>  **W1: Limited innovation**
>
> While prior works rely on **static prompt construction**, AE-CoT introduces a fundamentally different paradigm: **structured and optimizable reasoning search**.
>
> Specifically, our method differs from template engineering in three key aspects:
>
> - **Structured parameterization**: We decompose reasoning into **nine interpretable dimensions**, transforming CoT from unstructured text into a **compositional search space**. This enables systematic exploration rather than manual prompt design.
> - **Search-based optimization**: Instead of relying on handcrafted templates, we formulate jailbreak as an **optimization problem over reasoning structures**, solved via evolutionary search with crossover and mutation at the fragment level.
> - **Adaptive control mechanism**: We introduce a **fitness-driven adaptive mutation strategy**, allowing the search process to dynamically balance exploration and exploitation.
>
> Therefore, AE-CoT is not simply “cue engineering,” but rather a **general optimization framework over structured reasoning**, where templates (including those derived from prior work such as H-CoT) serve only as initialization.
>
> This distinction is analogous to the shift from:
> 1. static prompt design → prompt optimization,
> 2. heuristic construction → search-based methods.
>
> Importantly, our empirical results show that AE-CoT consistently outperforms template-based baselines, indicating that the performance gains stem from the **optimization process itself**, rather than the underlying cues.
>
> We will revise the paper to better emphasize this methodological perspective.
>
> ---
>  **W2: Whether “inference is the attack surface”**
>
> We thank the reviewer for this insightful question. We agree that distinguishing whether vulnerabilities arise from inference trajectories or merely from longer outputs is important.
>
> Recent studies have increasingly shown that **inference-time reasoning itself introduces new attack surfaces**, beyond simple output length effects. For example:
> - Prior work [1] shows that **chain-of-thought reasoning exposes intermediate semantic structures**, which can be manipulated to induce unsafe behavior.
> - Other studies [2] demonstrate that attacks can be **embedded within reasoning processes**, rather than only in the input prompt.
> - More recent work [3] further indicates that **multi-step reasoning increases model vulnerability**, as errors or biases can propagate across intermediate steps.
>
> These findings consistently suggest that the vulnerability is not merely due to longer or more detailed outputs, but due to the **structured reasoning trajectory itself**, which introduces additional degrees of freedom for adversarial manipulation.
>
> Our method directly builds on this insight by:
> - explicitly modeling reasoning as a **structured, multi-dimensional space**, and
> - systematically searching over **reasoning trajectories**, rather than surface-level prompts.
>
> Therefore, AE-CoT does not simply exploit verbosity, but rather **leverages the inherent vulnerability of inference-time reasoning processes**, which has been independently observed in multiple prior works.
>
> We will revise the paper to better clarify this distinction and strengthen the connection to existing literature.
>
> ---
>
> [1] Think in Safety: Unveiling and Mitigating Safety Alignment Collapse in Multimodal Large Reasoning Model
>
> [2] Chain-of-Thought Hijacking
>
> [3] Towards Safer Large Reasoning Models by Promoting Safety Decision-Making before Chain-of-Thought Generation
>
> ---
> **W3: Reliance on LLM-based evaluation**
>
> We thank the reviewer for this important suggestion. To address potential bias in LLM-based evaluation, we conducted additional **human evaluation experiments**.
>
> We recruit 5 independent annotators. For each response gained from AdvBench subset, if ≥3 annotators agree that the jailbreak is successful, it is counted as a success.
>
> **Results (Human ASR):**
>
> | Model | Human ASR |
> |-|-|
> |o1-mini|88%|
> | o3-mini | 84% |
> | GPT-5 | 48% |
> | DeepSeek-R1 | 90% |
> | Qwen3 | 88% |
> | Gemini-2.5-Flash | 94% |
>
> We observe that:
> - Human evaluation **confirms high attack success rates across most models**, consistent with our main results.
> - While absolute values are slightly lower than LLM-based evaluation (as expected due to stricter human judgment), the **overall trends remain consistent**.
> - Notably, stronger models (e.g., GPT-5) exhibit lower ASR under both evaluation protocols, indicating that the results reflect **model robustness rather than evaluation bias**.
>
> These findings demonstrate that our conclusions are **robust across both automated and human evaluation**, alleviating concerns about reliance on LLM judges.
>
> We will include these results in the revision to strengthen the evaluation.

---

> > ### Author Rebuttal · Reviewer_YgFw · 2026-04-03
> >
> > Thank you for the detailed rebuttal. I have raised my score to 4.

---

> > > ### Author Response · Authors · 2026-04-03
> > >
> > > We sincerely thank the reviewer for the encouraging feedback and thoughtful comments.
> > >
> > > We appreciate your careful consideration of our rebuttal and are glad that the additional experiments and clarifications have addressed your concerns. We are especially grateful for your reassessment and the updated score.
> > >
> > > Thank you again for your valuable feedback, which has contributed to improving the clarity and overall quality of our work.

---

### Official Review · Reviewer_nAmY · 2026-03-13

**Soundness:** 3
**Presentation:** 3
**Significance:** 3
**Originality:** 3
**Overall Recommendation:** 5
**Confidence:** 4

**Summary:**

This paper decomposed harmful goals into a nine-dimensional structured search space of interpretable reasoning fragments  then applying a genetic algorithm over this space to discover effective adversarial CoT suffixes, which can adapt to a model's refusal behavior

**Compliance With Llm Reviewing Policy:**

Affirmed.

**Final Justification:**

The authors's rebuttal responses have resolved my concern.

**Key Questions For Authors:**

Q1. The mutation rate adaptation rule is a simple with steps clipped to [0.1, 0.3]. Pls provide with the comparion against other simple schedules (e.g., cosine annealing, a fixed rate of 0.2, or random restarts) ?
Q2. Pls provide the experimental results with other models except gpt-4o

**Strengths And Weaknesses:**

1)Strengths :
Well-motivated.
Efficiency up to 3x speedup.
Covered six frontier reasoning models and five non-reasoning models.
2)Weaknesses:
The search space is built by implementing H-CoT against OpenAI-o1 and extracting interpretable attributes from the resulting traces. This means the novelty of this method is partly inherited from H-CoT.
The rule for  mutation rate is simple: decreased by 0.1 when fitness improves, increase by 0.1 when it doesn't.
The impact statement is insufficient for an attack paper achieving these results.

---

> ### Author Rebuttal · Authors · 2026-03-30
>
> ###
> We sincerely thank the reviewer for the positive evaluation and constructive feedback. Below we respond to the reviews in Weaknesses (W) and Questions (Q).
>
> ---
>
> **W1: Novelty and relation to H-CoT**
>
> We appreciate the reviewer’s concern regarding the relation to H-CoT. We clarify that while our structured search space is **inspired by reasoning traces obtained via H-CoT**, AE-CoT differs fundamentally in both **representation, search space construction, and optimization paradigm**.
>
> - **From static templates to parameterized reasoning space.**
>   H-CoT provides *static prompting templates* derived from demonstrations, which are applied in a one-shot manner. In contrast, AE-CoT transforms these traces into a **structured and optimizable parameter space (Θ)** by decomposing them into attribute-level components. This allows the method to move from direct reuse of prompts to **systematic exploration over a structured space**.
>
> - **From holistic prompts to compositional reasoning structure.**
>   Instead of treating CoT as monolithic text, we decompose it into **nine interpretable dimensions**, each with a candidate pool. This enables **cross-dimensional recombination**, producing a large combinatorial space of reasoning strategies that goes beyond the original H-CoT traces. As a result, AE-CoT can generate **novel reasoning patterns**, rather than replicating existing ones.
>
> - **From single-pass prompting to iterative, feedback-driven optimization.**
>   H-CoT relies on a fixed prompting pipeline, whereas AE-CoT introduces an **adaptive evolutionary framework** with fragment-level crossover and mutation. The search is guided by a scoring model and evolves over multiple generations, enabling **progressive refinement and exploration–exploitation trade-offs**.
>
> Importantly, AE-CoT consistently **outperforms H-CoT across models** (Table 1–2), demonstrating that the performance gains stem from the **structured optimization process**, rather than the source of initialization.
>
> More broadly, our approach follows a common paradigm in machine learning: leveraging existing traces as a foundation, while the **primary contribution lies in how they are parameterized, recombined, and optimized**. This distinguishes AE-CoT from prior template-based or demonstration-based methods.
>
> ---
> **W2 & Q1: Mutation rate design and comparison with alternatives**
>
> We agree that the mutation rule is simple, and emphasize that this is an intentional design choice. In black-box, query-limited settings, overly complex schedules may introduce instability without reliable feedback. Our key contribution is the **fitness-driven adaptive mechanism**, which adjusts exploration based on search progress.
>
> Following the reviewer’s suggestion, we compare our method with fixed (μ = 0.2), cosine annealing, and random mutation on AdvBench subset (Gemini-2.5-Flash-Thinking):
>
> | Method   | ASR ↑ | Avg Time (s) ↓ |
> |----------|------|----------------|
> | Adaptive | **100%** | **206.3** |
> | Fixed    | 90%  | 295.7 |
> | Cosine   | 90%  | 288.8 |
> | Random   | 88%  | 562.0 |
>
> We observe that:
> - **Adaptive mutation achieves the highest ASR (100%)**, outperforming all baselines.
> - It is also **more efficient**, ~1.4× faster than fixed and cosine schedules.
> - Random mutation yields both lower ASR and slower convergence.
>
> These results show that our approach provides a better **exploration–exploitation trade-off**, leading to both higher effectiveness and faster convergence.
>
> ---
> **W3: Insufficient Impact statement**
>
> We thank the reviewer for highlighting this important point. We will expand the impact statement to more clearly address responsible use.
>
> Specifically, we will:
> - clarify that all experiments are conducted in **controlled benchmark settings**,
> - position AE-CoT as a **red-teaming tool** for testing model safety,
> - discuss future directions on **process-level defense mechanisms**, such as monitoring intermediate reasoning steps.
>
> ---
>
> **Q2: Evaluation on models beyond GPT-4o**
>
> We thank the reviewer for this question.
> In our original setup, we use GPT-4o as a fixed judge for consistency. Following the reviewer’s suggestion, we additionally evaluate multiple different judge models on a subset of AdvBench (attacking o3-mini). The results are summarized below:
>
> | Judge Model | ASR ↑ | HS ↑ |
> |-------------|------|------|
> | GPT-4o      | 90%  | 86   |
> | Qwen-Max    | 80%  | 74   |
> | Grok-3      | 100% | 76  |
> | GPT-5       | 100% | 88   |
>
> We observe that while absolute scores vary across judge models (due to differences in safety alignment), the overall conclusions remain consistent:
> - AE-CoT achieves **high attack success rates across all judges**,
> - and maintains **strong harmfulness scores** under different evaluation criteria.
>
> These results suggest that our findings are **robust to the choice of judge model**, and do not depend on a specific evaluator. We will include these additional analyses in the revision to improve clarity.

---

> > ### Author Rebuttal · Reviewer_nAmY · 2026-04-06
> >
> > Thank you for the helpful responses. I have updated my score to acceptance.

---

> > > ### Author Response · Authors · 2026-04-06
> > >
> > > We sincerely thank the reviewer for the supportive feedback and thoughtful comments.
> > >
> > > We appreciate your careful review and are glad that our responses have addressed your concerns. We are grateful for your reconsideration of our work and for updating your score to acceptance.
> > >
> > > Thank you again for your valuable feedback, which has helped improve the clarity and strength of our paper.

---

### Decision · Program_Chairs · 2026-04-30

**Decision:**

Accept (regular)

**Comment:**

This paper uses the reasoning traces of LRMs as an attack surface. They propose AE-CoT, which first rewrites harmful goals into teacher-style educational requests, then decomposes a CoT jailbreak template into nine interpretable attributes, and finally performs multi-generation evolutionary search over this structured space with an adaptive mutation-rate mechanism. All the reviewers recognise the method's effectiveness and contribution, and all of them give a positive score to this paper. Therefore, I recommend accepting the paper.